# Paediatric Histoplasmosis 2000–2019: A Review of 83 Cases

**DOI:** 10.3390/jof7060448

**Published:** 2021-06-04

**Authors:** Rebecca MacInnes, Adilia Warris

**Affiliations:** 1Aberdeen Fungal Group, Institute of Medical Sciences, University of Aberdeen, Aberdeen AB25 2ZD, UK; rebecca.macinnes.15@abdn.ac.uk; 2MRC Centre for Medical Mycology, University of Exeter, Geoffrey Pope Building, Stocker Road, Exeter 4EX 4QD, UK

**Keywords:** histoplasmosis, disseminated disease, children, infants, treatment, *Histoplasma* spp.

## Abstract

Histoplasmosis is an endemic fungal infection that is confined to specific geographical regions. *Histoplasma* spp. are primary pathogens that cause disease in both immunocompetent and immunocompromised patients, ranging from a single-organ (mostly affecting the lungs) infection to life-threatening disseminated disease. Knowledge about the clinical epidemiology relies on data from adult populations; little is known about the patient and disease characteristics in the paediatric population. Therefore, a structured review of published cases of paediatric histoplasmosis between 2000 and 2019 was performed. A literature search of PubMed was conducted and the epidemiological and clinical data from 83 cases were analysed. The mean age at presentation was 9.5 ± 5.5 years, and 51% were girls. Two-thirds of the children were immunocompromised. The majority of children presented with disseminated disease. The most frequently observed clinical symptoms were respiratory symptoms, alongside non-specific systemic features, including fever, myalgia, fatigue and weight loss. The mortality rate was 11%. Histoplasmosis affects children of any age. Being immunocompromised is a risk factor for severe and disseminated disease. The lack of specific presenting features leads to underreporting and delay in diagnosis. To improve the recognition and outcome of histoplasmosis in childhood, increased awareness and surveillance systems are warranted.

## 1. Introduction

Histoplasmosis was first described in a paediatric patient in 1934, almost 30 years after the first adult patient with fatal disseminated histoplasmosis was described by Samuel Darling in 1906 [1,2]. Histoplasmosis is caused by the dimorphic fungus *Histoplasma capsulatum*, which can be divided into two varieties: *H. capsulatum* var. *capsulatum* and *H. capsulatum* var. *duboisii* [3]. *H. capsulatum* var. *duboisii* is found in Africa, whereas *H. capsulatum* var. *capsulatum* is most prevalent in the other endemic regions [4]. In addition to the traditional endemic regions, including the Ohio and Mississippi River Valleys in the United States, Central and South America and Africa, the true global distribution of histoplasmosis has recently been reported to be much more widespread [5]. Seroprevalence data and epidemiological studies from South-East Asia (in particular Thailand), Queensland and New South Wales in Australia, the Po Valley area in Italy, Quebec and Ontaria in Canada and an increased number of states in the United States show that histoplasmosis is endemic across the world [5]. *Histoplasma* conidia are found in these geographic areas in soil contaminated with bird and bat guano. Activities disturbing the soil lead to the release of the infectious conidia into the air, with infection acquired by inhaling contaminated air.

Every year, thousands of people in North and Central America and Africa are infected with *Histoplasma* spp.; however, most healthy individuals remain asymptomatic, as shown by an outbreak that occurred in Iowa in 1964 [6]. Six thousand people had asymptomatic infections whilst only 87 had pulmonary disease and 1 had disseminated disease. Immunocompromised patients are at risk of developing more severe primary infections, as well as reactivation of latent histoplasmosis and progression to disseminated disease, especially in patients with HIV [7].

Histoplasmosis in children can be asymptomatic or present as acute pulmonary histoplasmosis or disseminated histoplasmosis. Isolated single-organ infections are rarely reported [3,4]. Acute pulmonary histoplasmosis tends to be self-limiting and presents with fever, headache, malaise, chest pain and cough [8]. Presenting signs and symptoms of disseminated histoplasmosis are fever, malaise, anorexia and weight loss, hepatosplenomegaly, lymphadenopathy, mucous membrane ulceration and skin lesions [9,10,11]. Clinical features of histoplasmosis are non-specific and can mimic malignancies or tuberculosis [12,13].

In children living in endemic areas, asymptomatic histoplasmosis is common [14]. However, infants and immunocompromised paediatric patients are at higher risk of developing more severe disease [4,15]. Most knowledge of paediatric histoplasmosis is derived from outbreaks and case reports, and clinical management is based on adult data and experience [16]. This gap in knowledge increases the risk of late diagnosis or initial misdiagnosis, which can lead to more severe disease, and therefore, significant morbidity and mortality for children living in endemic areas. The aim of this study was to review all reported unique paediatric cases of histoplasmosis from 2000 to 2019 and extract data regarding the patient demographics, clinical features, diagnostic modalities, treatment and outcome to provide a comprehensive overview of childhood histoplasmosis.

## 2. Materials and Methods

A systematic literature search was conducted using PubMed. The following search terms: (histoplasmosis) OR (*Histoplasma*) were used. The following limits were applied to the search: “humans”, “age: 0–18 years” and “published between 1 January 2000–1 January 2019”. Cases were included in accordance with the inclusion and exclusion criteria (Table 1). This was done first through title analyses, then abstract analyses. Lastly, full-text analyses of the remaining results were conducted to produce a final set of studies included for final review. Diagnosis of histoplasmosis was defined as: identification of *Histoplasma* spp. via histopathological examination of tissue samples or culture or molecular methods and/or positive *Histoplasma* antigen and/or positive *Histoplasma* antibody detection. If the diagnostic modality was not specified, cases were included only if a definitive diagnosis was disclosed. Cases that did not disclose diagnostic information and specified the diagnosis as “probable” were excluded. Additionally, cases in which the age was not specified were excluded. In order to prevent selection bias, the entire search and selection process was repeated by another independent assessor (A.W.). Any inconsistencies were discussed until there was an agreement as to what studies should be included.

For each case, the following data were recorded: age, gender, disease type (single organ vs. disseminated disease), sites of infection, comorbidities, clinical features, diagnostic test results, treatment and patient outcome. When data on the infection sites were not explicitly provided, clinical features, radiographic findings or specimens used to identify the presence of *Histoplasma* spp. were used to define this. Data were managed in Microsoft Excel^®^ and descriptive statistics were used to present the data. Statistically significant results were defined as *p* < 0.05. As the sample sizes were small, when assessing the clinical significance of two nominal variables, Fisher’s exact test was used.

## 3. Results

Our search strategy yielded a total of 309 papers for histoplasmosis, of which 244 papers were excluded and 65 studies were included in this review. The process of inclusion and exclusion is outlined in Figure 1. Out of the 65 papers included in the case analysis, 83 cases were identified [17,18,19,20,21,22,23,24,25,26,27,28,29,30,31,32,33,34,35,36,37,38,39,40,41,42,43,44,45,46,47,48,49,50,51,52,53,54,55,56,57,58,59,60,61,62,63,64,65,66,67,68,69,70,71,72,73,74,75,76,77,78,79,80,81]. The details of the individual cases included in this review are presented in Appendix A.

### 3.1. Patient Characteristics

The mean age (±standard deviation) at presentation was 9.5 ± 5.5 years, with a median age of 11 years; range: 1 month–17 years (Table 2). Of the 83 patients, 13 (15.6%) were infants (≤2 years). Gender was reported in 82 cases, of which 42 were female (51.2%). Most cases were reported from North America (63.9%), followed by Asia (16.9%), South America (12%) and Africa (6%). Of the 53 cases presented in North America, one young infant was diagnosed in a non-endemic area of the United States. Infection was most likely acquired via vertical transmission, with the mother being previously infected in Guatemala. The one child reported in Europe was thought to have acquired the infection in the Democratic Republic of Congo.

The patient’s health status was specified in 70 cases, and 49 (72.5%) had an underlying condition reported, resulting in some form of immunosuppression (Table 2). Primary immunodeficiencies, including the hyper-IgE syndrome (*n* = 4), STAT-1 mutation (*n* = 2), T-cell deficiency (*n* = 1), interferon-gamma-receptor-1 deficiency (*n* = 1), common variable immunodeficiency (*n* = 1) and hemophagocytic lymphohistiocytosis (*n* = 1), as well as HIV, were the most common cause of immunosuppression (20.4% and 18.4%, respectively), followed by a history of renal transplantation (16.3%).

### 3.2. Disease Characteristics

Disease characteristics could be retrieved from 78 out of 83 cases (94%) (Table 3). Of these, 62 presented with a disseminated form of histoplasmosis, 12 had isolated pulmonary histoplasmosis, 2 had an isolated central nervous system infection, 1 suffered from laryngeal histoplasmosis and 1 had an isolated bone infection. The most commonly infected sites were the lungs (55.8%), followed by the lymphatic system (39%) and bone marrow (31.2%) (Table 3). No differences in the infection sites or types of diseases were observed when comparing immunocompromised to immunocompetent patients (*p* > 0.05, Fisher’s exact test).

### 3.3. Clinical Disease Presentation

Data on the presented clinical signs and symptoms was available for 78 out of 83 cases (94%). Fever was the most commonly presented feature, occurring in 53 cases (67.9%), followed by cough (29.5%) and weight loss (28.2%) (Figure 2). Less frequent signs and symptoms included (≤10%): nausea/vomiting (*n* = 8), diarrhoea (*n* = 7), bone/joint pain (*n* = 7), anorexia (*n* = 6), chest pain (*n* = 5), night sweats (*n* = 5), bone/joint swelling (*n* = 4), myalgia (*n* = 4), dysphagia/odynophagia (*n* = 4), seizures (*n* = 3) and slurred speech (*n* = 2). In one case, the diagnosis of histoplasmosis was an accidental finding.

### 3.4. Diagnosis

Details of the diagnostic modalities used were available for 94% of the paediatric patients. Histopathology or direct demonstration of *Histoplasma* spp. confirmed the diagnosis in 44 (56.4%) children, whilst a positive culture was obtained in 15 (19.2%) children. The identification of specific antibodies was performed in 37 cases, either via immunodiffusion (*n* = 7), complement fixation (*n* = 8) or a combination of both (*n* = 10). In 12 patients, the method of antibody detection was not mentioned. *Histoplasma* antigen detection was performed in 35 cases. Molecular methods were used to confirm the presence of *Histoplasma* spp. in two cases. Table 4 provides a detailed overview of which samples were used for the diagnosis.

### 3.5. Treatment and Outcome

Treatment details were retrieved for 69 (83%) patients (Table 5). Seven children did not receive antifungal treatment, and for six children, treatment was not specified. The most common antifungal regimen was an initial treatment with amphotericin B, followed by itraconazole, followed by an itraconazole treatment, especially in those with isolated pulmonary infection (50%). Alternative azoles used included ketoconazole, fluconazole, voriconazole and posaconazole. The mean (± standard deviation) duration of the antifungal treatment was 9.0 ± 7.4 months. Outcome data were available for 73 children (Table 3 and Table 5). The majority of patients were successfully treated (*n* = 56, 76.7%). The mortality rate was 11.0% (*n* = 8). No differences in mortality were observed between the immunocompromised and immunocompetent paediatric patients (*p* > 0.05 using Fisher’s exact test).

## 4. Discussion

Our systematic and extensive review of the literature produced over a period of 20 years resulted in the identification of 83 unique cases of paediatric histoplasmosis fulfilling the criteria to be included in this review. Our review complements the few data available that were derived from single-centre or single-country clinical studies in the United States and Latin America [15,82,83,84].

The age range and gender distribution were comparable to the previously published data and show that children of all ages are susceptible to developing symptomatic histoplasmosis [82,83,84,85,86]. Infants are considered to be at a higher risk of developing a more severe and disseminated form of the disease [15]. All the 13 infants included in our study presented with disseminated disease, except one; however, the outcomes were comparable between infants and children aged >1 year in our study (10% vs. 11%).

The majority of the children included in this review were living in North America (63.9%). Histoplasmosis is the most common endemic mycosis in North and Central America and was initially believed to be geographically restricted to these areas. More recently, the presence of *Histoplasma* spp. in other geographic areas, including regions in Africa, Asia and South America, is appreciated and clinical disease is increasingly being recognised [18,87,88,89,90,91,92,93,94]. Our review seems to indicate that there is an underreporting of paediatric histoplasmosis from Africa (only 6% of the cases). Two recent reviews focusing on African histoplasmosis reported that 25% and 30% of the 57 and 94 cases, respectively, were children [95,96]. Lack of awareness and clinical recognition, as well as a lack of management resources, might well explain this observation. *Histoplasma* antigen testing is not available in many endemic areas [97], resulting in under-diagnosis and -reporting. Cases that presented outside the endemic areas are considered as “import diseases”, as illustrated by two paediatric patients included in our study [55,67]. Histoplasmosis was reported in Europe as a consequence of international travel and migration [3].

Disseminated disease was the most common presentation of histoplasmosis in our study (79.5%), followed by isolated pulmonary histoplasmosis (15.3%). The high number of paediatric patients that presented with disseminated disease is most likely not a reflection of the actual clinical epidemiology of histoplasmosis in infants and children. Publication bias and the high number of children being immunocompromised that were included in this review may have resulted in this remarkable observation. Other studies have reported lower rates of disseminated disease in non-selected paediatric patient cohorts, ranging between 29% and 64% [82,84]. Although a higher number of immunocompromised children included in our review suffered from disseminated disease (87.8% vs. 69.6%), this was not a statistically significant difference.

A high percentage (66.7%) of the children in our study suffered from an underlying condition that resulted in them being immunocompromised. HIV/AIDS, primary immunodeficiency and the receipt of a renal transplant were the most commonly reported co-morbidities. HIV/AIDS is a well-known risk factor for developing histoplasmosis [98,99] and was one of the main underlying conditions reported in our study. All children infected with HIV (*n* = 9, 12.9%) presented with disseminated disease. Histoplasmosis in kidney transplant patients carries high morbidity and mortality [100,101]. Of the nine children who developed histoplasmosis after kidney transplantation, all but one presented with disseminated disease and one child died. Unfortunately, this diagnosis was only made at the autopsy [60].

An underlying primary immunodeficiency was present in 10 children, rendering them susceptible to developing severe and disseminated histoplasmosis [23,25,29,38,41,44,59,71]. The importance of the IL-12/IFN-γ pathway and Th17-mediated responses in antifungal immune defenses is well acknowledged [102]. An increasing number of both paediatric and adult patients with inborn errors in those pathways have been reported, including patients with CD40L deficiency, IL12 or IFN-γ receptor deficiency and defects in STAT1 or STAT3 [103]. It is remarkable that the presentation of gastro-intestinal histoplasmosis mimics Crohn’s disease in patients with hyper IgE syndrome [103,104,105]. As an invasive fungal disease can be a presenting symptom of primary immunodeficiency, an immunological workup, including an HIV test, should be performed in a child diagnosed with severe and/or disseminated histoplasmosis.

Fever, cough and weight loss were the most frequently reported symptoms, with lymphadenopathy and/or hepatosplenomegaly being common findings on physical examination. The non-specific signs and symptoms of histoplasmosis are well known, and therefore, the differential diagnosis can cover a wide range from common childhood pneumonia, to tuberculosis, to systemic diseases, including malignancies. Knowledge of the geographic distribution of *Histoplasma* spp. and recognising host factors that expose the patient to increased risk of developing histoplasmosis will be of value for enabling a timely diagnosis and management.

Isolated CNS infection occurred in two immunocompetent children [34,46]; however, 10.4% had some form of CNS involvement. This is in contrast with the observations done in 48 Colombian children with histoplasmosis in which CNS involvement was observed in 48% of those with disseminated histoplasmosis [82]. Delay in diagnosis and/or routine screening for CNS involvement might explain the observed differences. Involvement of the CNS in adult patients with histoplasmosis is a rare finding [106,107].

Proving a diagnosis of histoplasmosis includes demonstrating the presence of the fungus via histopathology or direct microscopy in a tissue sample from an affected site or recovery of the fungus via a culture in a specimen from a normally sterile site [108]. A proven diagnosis, as defined by the EORTC-MSG consensus definitions, was obtained in 75.6% of the paediatric patients presented in this review. *Histoplasma* antibody testing and antigen testing were performed to make a diagnosis in 47.4% and 44.9% of children, respectively. For the antibody testing, standard assays employing either complement fixation or immunodiffusion are available, with a reported sensitivity as high as 90% [109]. Paediatric-specific data about the performance of these assays are scarce, but two paediatric single-centre studies do suggest that a high percentage of children with histoplasmosis test antibody positive [82,84]. The same holds true for antigen testing, which has been used most commonly on urine samples with reported sensitivities between 76 and 90%. Sensitivity is lower when serum samples are used, with some good experiences reported using CSF and bronchoalveolar lavage samples aiding in the diagnostic process [109].

To treat paediatric histoplasmosis, the Infectious Disease Society of America (IDSA) guideline recommends following the treatment indications and regimens as used in adults [16]. Itraconazole can be used for mild-to-moderate disease, with amphotericin B (2–4 weeks) as the initial treatment for more severe and disseminated disease, followed by itraconazole for a total duration of 3 months. A longer duration of treatment is recommended depending on the severity of the histoplasmosis, concomitant use of immunosuppressive drugs or primary immunodeficiency. In our study, the majority of the patients (59%) received either itraconazole or a combination of amphotericin B followed by itraconazole. In addition, 11 (13%) patients received only amphotericin B, or amphotericin B followed by another azole (*n* = 9, 10.8%). The duration of treatment was remarkably long, with a mean of 9 months (+/−7.4 months). The high percentage of disseminated infections and the underlying immunocompromised conditions certainly influenced these prolonged treatment courses. No differences were noted in outcomes between immunocompromised and non-immunocompromised conditions and between the treatment regimens used. The mortality rate was 11% and comparable to observations amongst 40 infants with histoplasmosis [15], but higher than the 2.7% of 73 children with histoplasmosis reported by Quelletta et al. [84]. A study of 57 paediatric oncology patients with histoplasmosis showed no case fatalities [83], while a Colombian paediatric study could not report on mortality due to the retrospective nature of the study [82]. The higher mortality in our study is most likely explained by the higher number of children with immunosuppressive conditions, but a publication bias cannot be excluded.

To summarise, our literature review of 83 cases of paediatric histoplasmosis shows that especially children with underlying immunocompromised disorders are at risk for disseminated disease and present with aspecific signs and symptoms. Our review indicates that there is an underreporting of paediatric histoplasmosis in Africa, with a lack of awareness and clinical recognition, as well as a lack of management resources being the most likely explanations. In addition, the real clinical epidemiology of paediatric histoplasmosis is unknown, as mild disease and pulmonary histoplasmosis might go unnoticed and/or undiagnosed. Paediatric histoplasmosis is characterised by significant morbidity, with a fatal outcome in 11% of cases. Improved surveillance and awareness are warranted to obtain a better insight into the clinical epidemiology and to enable a timely diagnosis and treatment.

## Figures and Tables

**Figure 1 jof-07-00448-f001:**
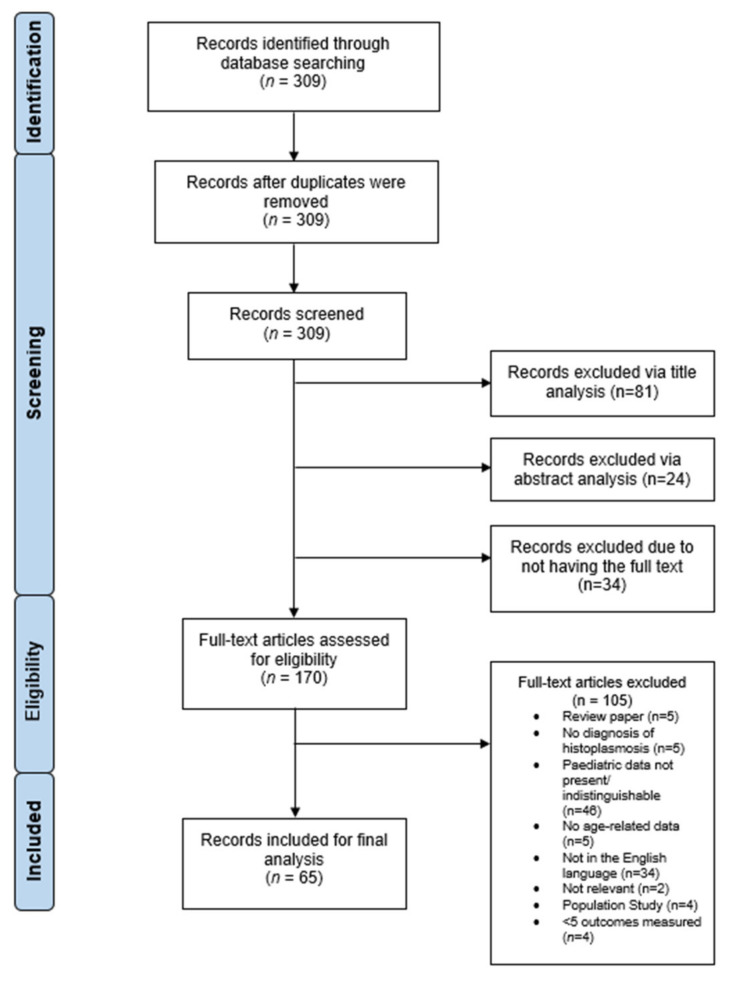
PRISMA flow chart demonstrating the process of inclusion/exclusion of histoplasmosis studies for analysis. The initial literature search found 309 papers related to histoplasmosis. Of these, 81 were excluded via title analysis, 24 were excluded via abstract analysis and 34 had no access to the full text. This left 170 studies, of which, 105 were excluded via full-text analysis; a total of 65 studies were included in the final analysis. Diagram adapted from Moher, D., Liberati, A., Tetzlaff, J., Altman, D.G., The PRISMA Group. (2009).

**Figure 2 jof-07-00448-f002:**
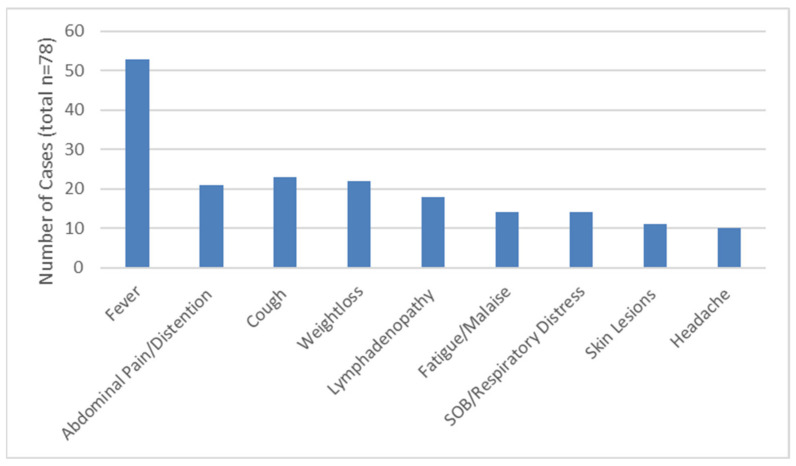
The most commonly presented features of the 78 children with histoplasmosis.

**Table 1 jof-07-00448-t001:** Inclusion and exclusion criteria that were used in the literature search.

	Inclusion Criteria	Exclusion Criteria
**Patient characteristics**	<18 years	Exclusively ≥18 years
	Diagnosis of histoplasmosis	No diagnosis of histoplasmosis
**Study**	Published 1 January 2000– 1 January 2019	Published before 1 January 2000
	Full text	No full text
	English language	Not in the English language
	Paediatric data distinguishable from adult data	Paediatric data indistinguishable from adult data
	Original case reports of histoplasmosis	No original case reports of histoplasmosis
		Population study in which the characteristics of individual cases were not described
		Systematic/literature reviews
**Outcome measures**	≥5 of the following outcomes described: age, gender, infection type, underlying conditions, presenting signs and symptoms, method(s) of diagnosis, treatment and patient outcome	<5 of the following outcomes described: age, gender, infection type, underlying conditions, presenting signs and symptoms, method(s) of diagnosis, treatment and patient outcome

**Table 2 jof-07-00448-t002:** Patient demographics and characteristics of 83 children with histoplasmosis.

Age (*n* = 83)	
Mean ± SD	9.5 ± 5.5 years
Median (range)	11 years (1 month–17 years)
Gender (*n* = 82)	
Female/male	42 (51.2%)/40 (48.8%)
Underlying condition (*n* = 70)	
None	21 (30.0%)
Primary immunodeficiency *	10 (14.3%)
HIV/AIDS	9 (12.9%)
Renal transplant	8 (11.4%)
Crohn’s disease	6 (8.6%)
Juvenile rheumatoid arthritis	6 (8.6%)
Haematological malignancy ^&^	4 (5.7%)
Other ^#^	5 (7.2%)
Disease Type (*n* = 78)	
Disseminated	62 (79.5%)
Pulmonary	12 (15.4%)
Central nervous system	2 (2.6%)
Other ^$^	2 (2.6%)
Geographic Area (*n* = 83)	
North America	53 (63.9%)
Asia	14 (16.9%)
Europe	1 (1.2%)
Africa	5 (6.0%)
South America	10 (12.0%)

* Hyper-IgE syndrome (*n* = 4), STAT1 mutation (*n* = 2), T-cell deficiency (*n* = 1), interferon-gamma-receptor-1 deficiency (*n* = 1), common variable immunodeficiency (*n* = 1), hemophagocytic lympho-histiocytosis (*n* = 1). ^&^ Acute lymphoblastic leukaemia (*n* = 2), acute myeloid leukaemia (*n* = 1), Hodgkin’s lymphoma (*n* = 1). ^#^ Idiopathic chronic anterior uveitis (*n* = 2) and juvenile systemic lupus erythematosus (*n* = 1) treated with immunosuppression; gastrointestinal stromal tumor (n = 1) and Wilms tumor (n = 1) treated with chemotherapy; EBV infection (*n* = 2); tuberculosis (*n* = 1). ^$^ Osseous (*n* = 1) and laryngeal (*n* = 1) infections.

**Table 3 jof-07-00448-t003:** Disease characteristics and outcome compared between immunocompetent and immunocompromised children with histoplasmosis.

	Total	Immunocompromised	Immunocompetent
Infection site	*n* = 78	*n* = 40	*n* = 23
Lungs	43 (55.8%)	26 (65%)	10 (43.5%)
Lymph nodes	30 (39%)	16 (40%)	7 (30.4%)
Bone marrow	24 (31.2%)	16 (40%)	6 (26.1%)
Skin	12 (15.6%)	7 (17.5%)	4 (17.4%)
Central nervous system	8 (10.4%)	2 (5%)	3 (13%)
Bone	6 (7.8%)	2 (5%)	2 (8.7%)
Other *	13 (16.9%)	12 (3%)	1 (4.3%)
Disease type	*n* = 78	*n* = 41	*n* = 23
Disseminated	62 (79.5%)	36 (87.8%)	16 (69.6%)
Single organ	16 (20.5%)	5 (12.2%)	7 (30.4%)
Lungs	12 (15.4%)	4 (9.8%)	4 (17.4%)
Central nervous system	2 (2.6%)	-	2 (8.7%)
Other	2 (2.6%)	1 (2.4%), larynx	1 (4.3%), bone
Outcome	*n* = 73	*n* = 43	*n* = 19
Cure	35 (47.9%)	20 (46.5%)	10 (52.6%)
Clinical improvement	21 (28.8%)	11 (25.6%)	7 (36.8%)
Recurrence	6 (8.2%)	4 (9.3%)	1 (5.3%)
Death	8 (11%)	5 (11.6%)	1 (5.3%)
Lost to follow-up	3 (4.1%)	3 (7%)	-

* Other infection sites included: pericardium (*n* = 1), gastrointestinal tract (*n* = 4), larynx (*n* = 1), soft tissue (*n* = 2), liver (*n* = 2), spleen (*n* = 1), pancreas (*n* = 1), eye (*n* = 1).

**Table 4 jof-07-00448-t004:** Tissue and body fluids that were used to diagnose histoplasmosis in 83 paediatric patients.

	Number of Samples	Antibody	Antigen	Culture	Histopathology	Microscopy
Urine	24	-	24	-	-	-
Blood/serum	51	37	13	4	-	
Bone marrow ^#^	16	-	-	5	14	-
Lymph node	15	-	-	5	13	-
Bronchoalveolar lavage fluid	6	-	-	1	-	5
Skin ^#^	7	-	-	3	5	-
Lung tissue	6	-	-	-	6	-
Cerebrospinal fluid	5	-	4	2	-	1
Bone	5	-	-	1	5	-
Gastrointestinal tissue	4	-	-	2	3	-
Other *	5	-	2	-	3	-
Not specified	11	-	10	-	1	-

* Larynx exsudate (*n* = 1), joint tissue (*n* = 2), pericardial fluid (*n* = 1), pleural fluid (*n* = 1), soft tissue (*n* = 1); ^#^ molecular methods were used on one bone marrow sample and one skin sample.

**Table 5 jof-07-00448-t005:** Treatment and outcome of 83 children with histoplasmosis.

Antifungal Therapy	Number of Patients	Cured or Clinical Improvement	Recurrence	Death	Lost to Follow-Up/Not Specified
Antifungal Therapy					
AmB → Itra	31	23 (74.2%)	2	-	6
Itra only	18	15 (83.3%)	1	-	2
AmB only	11	6 (72.7%)	-	5	-
AmB → other azoles	6	5 (83.3%)	1	-	-
Other azoles	3	2 (66.7%)	1	-	-
None	7	4 (57.1%)	-	3	-
Not specified	6	1 (16.7%)	1	-	4

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
