# Peer review of "Paediatric Histoplasmosis 2000–2019: A Review of 83 Cases"

_jof, 2021, doi:10.3390/jof7060448_

Round 1

Reviewer 1 Report

The paper is very interesting and well organized.

However, the informations included in references 82-85 are only discussed but not detailed in the table for the global evaluation of the present review, because the cited papers do not disclose all the required infomations as described in the "Material and Methods" paragraph. They treat large clinical case series (Lopez: 45 cases; Adderson: 57 cases; Odio: 40 cases; Ouellette: 73 cases) cumulatively well described. For this reason the present paper title could include the number of the evaluated cases (i.e. "Pediatric histoplasmosis: about 83 cases, 2000-2019" or similar).

I suggest to clarify the difference between "children" and "older children" (in "Discussion" they are separately treated, but the paragraph "Materials and Methods" paragraph does not specify the difference).

Author Response

1. We agree with the reviewer that the title could be more specific and we have modified as follow:

Pediatric histoplasmosis 2000-2019: a review of 83 cases.

2. We have to apologize for the confusion, and there is no difference between children and older children and no separate analyses were as such performed. We have deleted ‘older’ in line 190 (the only place where we used this terminology).

Reviewer 2 Report

Dear Authors, 

I had the great pleasure to review your article about histoplasmosis in pediatric patients. Overall I believe the article is very good and brings novelty to the current literature. Title however I think it should be more suggestive as well as the keywords section. 

Author Response

We thank the reviewer for her/his kind words of appreciation for our manuscript and have made the title more suggestive and have updated the keywords section by including ‘disseminated disease’, ‘infants’ and ‘treatment’.

Reviewer 3 Report

The authors review the pediatric cases of histoplasmosis published since 2010. The manuscript is well written with appropriate use of figures and tables. The introduction is of appropriate length, the methods section is clearly written and the results are presented well. The discussion is excellent, the authors explain there observations and discrepancies with previous case series and discuss the limitations of the review. 

I have only some small comments. 

  1. Line 30: "H. capsulatum var. duboisii is found in Africa [4], whereas H. capsula-30 tum var. capsulatum is most prevalent in North and Central America [5]." Histoplasma is far more widespread. Probably should also mentions other regions like SE Asia here too (like in the discussion).
  2. Materials and Methods: was molecular detection of Histoplasma not included in the definition diagnosis of histoplasmosis?
  3. The authors mention that histoplasmosis is underreported in Africa based on the low number of cases in literature. Is this statement well substantiated? Perhaps the number of cases is truly low in Africa. However, as 30% of H. capsulatum var duboisii cases were reported in Europe, it could be that many cases in Africa remain undetected (https://doi.org/10.1093/cid/ciaa1304). However, the authors should weaken this statement in the discussion and the final conclusion or add additional prove for the statement. . 

Author Response

1. We thank the reviewer for this comment and have now mentioned in the introduction that in SE Asia the Histoplasma capsulatum var. capsulatum is the most prevalent in.

2. Molecular diagnosis was included in the definition of diagnosis and histoplasmosis, and was used in only 2 cases. We have included this information in the Materials and Methods section as well as in table 4.

3. We thank the reviewer for this comment and we agree that it is not entirely clear if there is an underreporting of cases in Africa, but there is some recent data that suggests that there is indeed an underreporting. We have added an additional sentence and 2 references to support our statement, in the discussion.

Two recent reviews focusing on African histoplasmosis reported that 25% and 30% of the 57 and 94 cases, respectively, were children [96,97].